# QUANTITATIVE UNIVERSAL APPROXIMATION BOUNDS FOR DEEP BELIEF NETWORKS

## ABSTRACT

We show that deep belief networks with binary hidden units can approximate any multivariate probability density under very mild integrability requirements on the parental density of the visible nodes. The approximation is measured in the $L^q$-norm for $q \in [1, \infty]$ ($q = \infty$ corresponding to the supremum norm) and in Kullback-Leibler divergence. Furthermore, we establish sharp quantitative bounds on the approximation error in terms of the number of hidden units.

## 1 INTRODUCTION

Deep belief networks (DBNs) are a class of generative probabilistic models obtained by stacking several restricted Boltzmann machines (RBMs, Smolensky (1986)). For a brief introduction to RBMs and DBNs we refer the reader to the survey articles Fischer & Igel (2012; 2014); Montúfar (2016); Ghojogh et al. (2021). Since their introduction, see Hinton et al. (2006); Hinton & Salakhutdinov (2006), DBNs have been successfully applied to a variety of problems in the domains of natural language processing Hinton (2009); Jiang et al. (2018), bioinformatics Wang & Zeng (2013); Liang et al. (2014); Cao et al. (2016); Luo et al. (2019), financial markets Shen et al. (2015) and computer vision Abdel-Zaher & Eldeib (2016); Kamada & Ichimura (2016; 2019); Huang et al. (2019). However, our theoretical understanding of the class of continuous probability distributions, which can be approximated by them, is limited. The ability to approximate a broad class of probability distributions—usually referred to as *universal approximation property*—is still an open problem for DBNs with real-valued visible units. As a measure of proximity between two real-valued probability density functions, one typically considers the $L^q$-distance or the Kullback-Leibler divergence.

**Contributions.** In this article we study the approximation properties of deep belief networks for multivariate continuous probability distributions which have a density with respect to the Lebesgue measure. We show that, as $m \to \infty$, the universal approximation property holds for binary-binary DBNs with two hidden layers of sizes $m$ and $m + 1$, respectively. Furthermore, we provide an explicit quantitative bound on the approximation error in terms of $m$. More specifically, the main contributions of this article are:

- For each $q \in [1, \infty)$ we show that DBNs with two binary hidden layers and parental density $\varphi : \mathbb{R}^d \to \mathbb{R}_+$ can approximate any probability density $f : \mathbb{R}^d \to \mathbb{R}_+$ in the $L^q$-norm, solely under the condition that $f, \varphi \in L^q(\mathbb{R}^d)$, where

$$L^q(\mathbb{R}^d) = \left\{ f : \mathbb{R}^d \to \mathbb{R} : \|f\|_{L^q} = \left( \int_{\mathbb{R}^d} |f(x)|^q \, dx \right)^{\frac{1}{q}} < \infty \right\}.$$

  In addition, we prove that the error admits a bound of order $\mathcal{O}\big(m^{\frac{1}{\min(q,2)}-1}\big)$ for each $q \in (1, \infty)$, where $m$ is the number of hidden neurons.

- If the target density $f$ is uniformly continuous and the parental density $\varphi$ is bounded, we provide an approximation result in the $L^\infty$-norm (also known as supremum or uniform norm), where

$$L^\infty(\mathbb{R}^d) = \left\{ f : \mathbb{R}^d \to \mathbb{R} : \|f\|_{L^\infty} = \sup_{x \in \mathbb{R}^d} |f(x)| < \infty \right\}.$$

- Finally, we show that continuous target densities supported on a compact subset of $\mathbb{R}^d$ and uniformly bounded away from zero can be approximated by deep belief networks with bounded parental density in Kullback-Leibler divergence. The approximation error in this case is of order $\mathcal{O}\left(m^{-1}\right)$.

**Related works.** One of the first approximation results for deep belief networks is due to Sutskever & Hinton (2008) and states that any probability distribution on $\{0, 1\}^d$ can be learnt by a DBN with $3 \times 2^d$ hidden layers of size $d + 1$ each. This result was improved by Le Roux & Bengio (2010); Montúfar & Ay (2011) by reducing the number of layers to $\frac{2^{d-1}}{d-\log(d)}$ with $d$ hidden units each. These results, however, are limited to discrete probability distributions. Since most applications involve continuous probability distributions, Krause et al. (2013) considered Gaussian-binary DBNs and analyzed their approximation capabilities in Kullback-Leibler divergence, albeit without a rate. In addition, they only allow for target densities that can be written as an infinite mixture of a set of probability densities satisfying certain conditions, which appear to be hard to check in practice.

Similar questions have been studied for a variety of neural network architectures: The famous results of Cybenko (1989); Hornik et al. (1989) state that deterministic multi-layer feed-forward networks are universal approximators for a large class of Borel measurable functions, provided that they have at least one sufficiently large hidden layer. See also the articles Leshno et al. (1993); Chen & Chen (1995); Barron (1993); Burger & Neubauer (2001). Le Roux & Bengio (2008) proved the universal approximation property for RBMs and discrete target distributions. Montúfar & Morton (2015) established the universal approximation property for discrete restricted Boltzmann machines. Montúfar (2014) showed the universal approximation property for deep narrow Boltzmann machines. Montúfar (2015) showed that Markov kernels can be approximated by shallow stochastic feed-forward networks with exponentially many hidden units. Bengio & Delalleau (2011); Pascanu et al. (2014) studied the approximation properties of so-called deep architectures. Merkh & Montúfar (2019) investigated the approximation properties of stochastic feed-forward networks.

The recent work Johnson (2018) nicely complements the aforementioned results by obtaining an illustrative negative result: Deep narrow networks with hidden layer width at most equal to the input dimension do not posses the universal approximation property.

Since our methodology involves an approximation by a convex combination of probability densities, we refer the reader to the related works of Nguyen & McLachlan (2019); Nguyen et al. (2020) and the references therein for an overview of the wide range of universal approximation results in the context of mixture models. See also Everitt & Hand (1981); Titterington et al. (1985); McLachlan & Basford (1988); McLachlan & Peel (2000); Robert & Mengersen (2011); Celeux (2019) for in-depth treatments of mixture models.

The recent articles Bailey & Telgarsky (2018); Perekrestenko et al. (2020) in the context of generative networks show that deep neural networks can transform a one-dimensional uniform distribution in a way to approximate any two-dimensional Lipschitz continuous target density.

Another strand of research related to the questions of this article are works on quantile (or distribution) regression, see Koenker (2005) as well as Dabney et al. (2018); Tagasovska & Lopez-Paz (2019); Fakoor et al. (2021) for recent methods involving neural networks.

## 2 DEEP BELIEF NETWORKS

A restricted Boltzmann machine (RBM) is a an undirected, probabilistic, graphical model with bipartite vertices that are fully connected with the opposite class. To be more precise, we consider a simple graph $\mathcal{G} = (\mathcal{V}, \mathcal{E})$ for which the vertex set $\mathcal{V}$ can be partitioned into sets $V$ and $H$ such that the edge set is given by $\mathcal{E} = \left\{\{s, t\} : s \in V, t \in H\right\}$. We call vertices in $V$ *visible* units; $H$ contains the *hidden* units. To each of the visible units we associate the state space $\Omega_V$ and to the hidden ones we associate $\Omega_H$. We equip $\mathcal{G}$ with a *Gibbs probability measure*

$$\pi(v, h) = \frac{e^{-\mathscr{H}(v,h)}}{\mathcal{Z}}, \qquad v \in (\Omega_V)^V, h \in (\Omega_H)^H,$$

where $\mathscr{H} : (\Omega_V)^V \times (\Omega_H)^H \to \mathbb{R}$ is chosen such that $\mathcal{Z} = \iint e^{-\mathscr{H}(v,h)} \, dv \, dh < \infty$. Notice that the integral becomes a sum if $\Omega_V$ (resp. $\Omega_H$) is a discrete set. It is customary to identify the RBM with the probability measure $\pi$.

An important example are *binary-binary* RBMs. These are obtained by choosing $\Omega_V = \Omega_H = \{0, 1\}$ and

$$\mathscr{H} = \langle v, Wh \rangle + \langle v, b \rangle + \langle h, c \rangle, \quad v \in \{0, 1\}^V, \, h \in \{0, 1\}^H, \tag{1}$$

where $b \in \{0, 1\}^V$ and $c \in \{0, 1\}^H$ are called *biases*, and $W \in \mathbb{R}^{V \times H}$ is called the *weight matrix*. We shall write for $m, n \in \mathbb{N}$,

$$\text{B-RBM}(m, n) = \{\pi \text{ is a binary-binary RBM with } m \text{ visible and } n \text{ hidden units}\}, \tag{2}$$

for the set of binary-binary RBMs with fixed layer sizes.

The following discrete approximation result is well known, see also Montúfar & Ay (2011):

**Proposition 1** (Le Roux & Bengio (2008), Theorem 2). *Let $m \in \mathbb{N}$ and $\mu$ be a probability distribution on $\{0, 1\}^m$. Let*

$$\text{supp}(\mu) = \{v \in \{0, 1\}^m : \mu(v) > 0\}$$

*be the support of $\mu$. Set $n = |\text{supp}(\mu)| + 1$. Then, for each $\varepsilon > 0$, there is a $\pi \in \text{B-RBM}(m, n)$ such that*

$$\left| \mu(v) - \sum_{h \in \{0,1\}^n} \pi(v, h) \right| \leqslant \varepsilon \qquad \forall \, v \in \{0, 1\}^m.$$

A deep belief network (DBN) is constructed by stacking two RBMs. To be more precise, we now consider a tripartite graph with hidden layers $H_1$ and $H_2$ and visible units $V$. We assume that the edge set is now given by $\mathcal{E} = \{\{s, t_1\}, \{t_1, t_2\} : s \in V, t_1 \in H_1, t_2 \in H_2\}$. The state spaces are now $\Omega_V = \mathbb{R}$ and $\Omega_{H_1} = \Omega_{H_2} = \{0, 1\}$. We think of edges in the graph as dependence of the neurons (in the probabilistic sense). The topology of the graph hence shows that the vertices in $V$ and $H_2$ shall be conditionally independent, that is, we require that

$$p(v, h_1, h_2) = p(v \,|\, h_1) p(h_1, h_2). \tag{3}$$

The joint density of the hidden units $p(h_1, h_2)$ will be chosen as binary-binary RBM.

Let $\mathcal{D}(\mathbb{R}^d) = \{f : \mathbb{R}^d \to \mathbb{R}_+ : \int_{\mathbb{R}^d} f(x) \, dx = 1\}$ be the set of probability densities on $\mathbb{R}^d$. For $\varphi \in \mathcal{D}(\mathbb{R}^d)$ and $\sigma > 0$ we set

$$\mathcal{V}_\varphi^\sigma = \left\{ \varphi_{\mu,\sigma} = \sigma^{-d} \varphi \left( \frac{x - \mu}{\sigma} \right) : \mu \in \mathbb{R}^d \right\}. \tag{4}$$

Notice that all elements of $\mathcal{V}_\varphi^\sigma$ are themselves probability distributions. We fix a *parental density* $\varphi \in \mathcal{D}(\mathbb{R}^{|V|})$ and choose the conditional density in (3) as $p(\cdot \,|\, h_1) \in \mathcal{V}_\varphi^\sigma$ for each $h_1 \in H_1$.

**Example 2.** *The most popular choice of the parental function $\varphi$ in (4) is the $d$-dimensional standard Gaussian density*

$$\varphi(x) = \frac{1}{(2\pi)^{d/2}} \exp\left( -\frac{|x|^2}{2} \right), \qquad x \in \mathbb{R}^d. \tag{5}$$

*Another density considered in previous works is the truncated exponential distribution*

$$\varphi(x) = \prod_{i=1}^d \frac{\lambda_i e^{-\lambda_i x_i}}{1 - e^{-b_i \lambda_i}} \mathbb{1}_{[0, b_i]}(x_i), \qquad x = (x_1, \ldots, x_d) \in \mathbb{R}^d, \tag{6}$$

*where $b_i, \lambda_i > 0$ for each $i = 1, \ldots, d$.*

Similar to (2), we collect all DBNs in the set

$$\text{DBN}_\varphi(d, m, n) = \Big\{ p \text{ is a DBN with parental density } \varphi, \, d \text{ visible units}, \, m \text{ hidden}$$

$$\text{units on the first level, and } n \text{ hidden units on the second level} \Big\},$$

where $\varphi \in \mathcal{D}(\mathbb{R}^d)$ and $d, m, n \in \mathbb{N}$. We shall not distinguish between the whole DBN and the marginal density of the visible nodes, which is the object we are ultimately interested in, that is, we write

$$p(v) = \sum_{h_1 \in H_1} \sum_{h_2 \in H_2} p(v, h_1, h_2). \tag{7}$$

In case $p \in \mathsf{DBN}_\varphi(d, m, n)$ with $\varphi \in L^q(\mathbb{R}^{|V|})$ then also the marginal (7) belongs to $L^q(\mathbb{R}^{|V|})$.

After their introduction in Hinton & Salakhutdinov (2006), deep belief networks rose to prominence due to a training algorithm developed in Hinton et al. (2006) which addressed the vanishing gradient problem by pre-training deep networks. Instead of naïvely stacking two RBMs the authors considered several such stacked layers and greedily pre-trained the weights over the layers on a contrastive divergence loss. To be more precise, let $M$ denote the number hidden layers, then, first the visible and the first hidden layer are considered as a classical RBM and the weights of the first hidden layer are learnt. In the second step, the weights of the second hidden layer are learnt based on the first hidden layer using Gibbs sampling. This procedure repeats iteratively until all hidden layers are trained. For more details we refer to Fischer & Igel (2014); Ghojogh et al. (2021).

## 3 MAIN RESULTS

To state the results of this article, we need to introduce three bits of additional notation: Let $q \in [1, \infty]$. We declare $\mathcal{D}_q(\mathbb{R}^d) = \mathcal{D}(\mathbb{R}^d) \cap L^q(\mathbb{R}^d)$. Finally, for $q \in [1, \infty)$, let us abbreviate the constant

$$\Upsilon_q = \max\left(1, \frac{1}{\sqrt{2\pi}} \int_{-\infty}^{\infty} |x|^q e^{-\frac{x^2}{2}} \, dx\right)^{\frac{1}{q}} = \begin{cases} 1 & q \leqslant 2, \\ \dfrac{\sqrt{2}}{\pi^{\frac{1}{2q}}} \Gamma\left(\dfrac{q+1}{2}\right)^{\frac{1}{q}}, & q > 2, \end{cases} \tag{8}$$

with the Gamma function $\Gamma(x) = \int_0^\infty t^{x-1} e^{-t} \, dt$, $x > 0$.

The main results of this paper are stated in the following two theorems:

**Theorem 3.** *Let $q \in [1, \infty)$ and $f, \varphi \in \mathcal{D}_q(\mathbb{R}^d)$. Then, for each $m \in \mathbb{N}$, the following quantitative bound holds:*

$$\inf_{p \in \mathsf{DBN}_\varphi(d, m, m+1)} \|f - p\|_{L^q} \leqslant \frac{2\Upsilon_q \|\varphi\|_{L^q}}{m^{1 - \frac{1}{\min(q, 2)}}}, \tag{9}$$

*where the constant $\Upsilon_q$ is defined in (8).*

*While this bound becomes trivial if $q = 1$, the following qualitative approximation result still holds in that case: For any $\varepsilon > 0$, there is an $M \in \mathbb{N}$ such that, for each $m \geqslant M$, we can find a $p \in \mathsf{DBN}_\varphi(d, m, m+1)$ satisfying*

$$\|f - p\|_{L^q} \leqslant \varepsilon.$$

**Remark 4.** *Returning to Example 2, we find that $\|\varphi\|_{L^q} = q^{-\frac{d}{2q}}$ for the $d$-dimensional standard normal distribution (5) and*

$$\|\varphi\|_{L^q} = \prod_{i=1}^{d} \frac{\lambda_i^{1-\frac{1}{q}}}{q^{\frac{1}{q}}\left(1 - e^{-b_i \lambda_i}\right)} \left(1 - e^{-q\lambda_i b_i}\right)^{\frac{1}{q}}$$

*for the truncated exponential distribution (6). Our bound (9) thus shows that deep belief networks with truncated exponential parental density (for suitable choice of the parameters $b$ and $\lambda$) better approximate the target density $f$. This is especially prevalent for small $q$, which is the primary case of interest, see Corollary 7 below. For a detailed review of the exponential family's properties we refer to Brown (1986).*

To state the approximation in the $L^\infty$-norm, we need to introduce the space of bounded and uniformly continuous functions:

$$\mathcal{C}_u(\mathbb{R}^d) = \left\{ f \in L^\infty(\mathbb{R}^d) : \lim_{\delta \downarrow 0} \sup_{|x-y| \leqslant \delta} |f(x) - f(y)| = 0 \right\}.$$

Notice that any probability density $f \in \mathcal{D}(\mathbb{R}^d)$, which is differentiable and has a bounded derivative, belongs to $\mathcal{C}_u(\mathbb{R}^d)$ since any uniformly continuous and integrable function is bounded.

**Theorem 5.** *Let $f \in \mathcal{D}(\mathbb{R}^d) \cap \mathcal{C}_u(\mathbb{R}^d)$ and $\varphi \in \mathcal{D}_\infty(\mathbb{R}^d)$. Then, for any $\varepsilon > 0$, there is an $M \in \mathbb{N}$ such that, for each $m \geqslant M$, we can find a $p \in \mathsf{DBN}_\varphi(d, m, m+1)$ satisfying*

$$\|f - p\|_{L^\infty} \leqslant \varepsilon.$$

**Remark 6.** *The uniform continuity requirement on $f$ in Theorem 5 can actually be relaxed to essential uniform continuity, that is, $f$ is uniformly continuous except on a set with zero Lebesgue measure. The most notable example of such a function is the uniform distribution $f = \mathbb{1}_{[0,1]}$.*

Another important metric between between probability densities $p, q : \mathbb{R}^d \to \mathbb{R}_+$ is the *Kullback-Leibler divergence* (or *relative entropy*) defined by

$$\mathrm{KL}(f \| g) = \int_{\mathbb{R}^d} f(x) \log \left( \frac{f(x)}{g(x)} \right) \, dx$$

if $\{x \in \mathbb{R}^d : g(x) = 0\} \subset \{x \in \mathbb{R}^d : f(x) = 0\}$ and $\mathrm{KL}(f \| g) = \infty$ otherwise. From Theorems 3 and 5 we can deduce the following quantitative approximation bound in the Kullback-Leibler divergence:

**Corollary 7.** *Let $\varphi \in \mathcal{D}_\infty(\mathbb{R}^d)$. Let $\Omega \subset \mathbb{R}^d$ be a compact set and $f : \Omega \to \mathbb{R}_+$ be a continuous probability density. Suppose that there is an $\eta > 0$ such that both $f \geqslant \eta$ and $\varphi \geqslant \eta$ on $\Omega$. Then there is a constant $M > 0$ such that, for each $m \in \mathbb{N}$, it holds that*

$$\inf_{p \in \mathsf{DBN}_\varphi(d, m, m+1)} \mathrm{KL}(f \| p) \leqslant \frac{M}{\eta m} \left( 8\|\varphi\|_{L^2}^2 + \|f - \varphi\|_{L^2(\Omega)}^2 \right), \tag{10}$$

*where $\|f - \varphi\|_{L^2(\Omega)}^2 = \int_\Omega |f(x) - \varphi(x)|^2 \, dx$.*

Let us note that any $\varphi \in \mathcal{D}_\infty(\mathbb{R}^d)$ is square-integrable so that the right-hand side of the bound (10) is actually finite. This follows from the *interpolation inequality*

$$\|\varphi\|_{L^2} \leqslant \sqrt{\|\varphi\|_{L^1} \|\varphi\|_{L^\infty}} = \sqrt{\|\varphi\|_{L^\infty}}, \tag{11}$$

see (Brezis, 2011, Exercise 4.4).

**Remark 8.** *The first assertion of Theorem 3 and Theorem 5 generalize to deep belief networks with additional hidden layers, however, it is still an open question whether* (9) *can be improved by adding more depth, see also Jalali et al. (2019) for an analysis of this question in the context of Gaussian mixture models.*

Corollary 7 considerably generalizes the results of (Krause et al., 2013, Theorem 7). There, the authors only prove that deep belief networks can approximate any density in the closure of the convex hull of a set of probability densities satisfying certain conditions, which appear to be difficult to check in practice. That work also does not contain a convergence rate. In comparison, our results directly describe the class of admissible target densities and do not rely on the indirect description through the convex hull. Finally, there is an unjustified step in the argument of Krause et al., which appears hard to reconcile, see Remark 16 below for details.

## 4 PROOFS

This section presents the proofs of Theorems 3, 5 and Corollary 7. As a first step, we shall establish a couple of preliminary results in the next two subsections.

### 4.1 $L^q$-APPROXIMATION OF FINITE MIXTURES

Given a set $A \subset L^q(\mathbb{R}^d)$, the *convex hull* of $A$ is by definition the smallest convex set containing $A$; in symbols $\mathrm{conv}(A)$. It can be shown that

$$\mathrm{conv}(A) = \left\{ \sum_{i=1}^n \alpha_i a_i : \alpha = (\alpha_1, \ldots, \alpha_n) \in \triangle_n, a_1, \ldots, a_n \in A, n \in \mathbb{N} \right\}$$

with $\triangle_n = \left\{ x \in [0,1]^n : \sum_{i=1}^n x_i = 1 \right\}$, the $n$-dimensional standard simplex. It is also convenient to introduce the *truncated convex hull*

$$\mathrm{conv}_m(A) = \left\{ \sum_{i=1}^m \alpha_i a_i : \alpha = (\alpha_1, \ldots, \alpha_m) \in \triangle_m, a_1, \ldots, a_m \in A \right\}$$

for $m \in \mathbb{N}$ so that $\mathrm{conv}(A) = \bigcup_{m \in \mathbb{N}} \mathrm{conv}_m(A)$. The *closed convex hull* $\overline{\mathrm{conv}}(A)$ is the smallest closed convex set containing $A$ and it is straight-forward to check that it coincides with the closure of $\mathrm{conv}(A)$ in the topology of $L^q(\mathbb{R}^d)$.

The next result shows that we can approximate any probability density in the truncated convex hull of the set (4) arbitrarily well by a DBN with a fixed number of hidden units:

**Lemma 9.** *Let* $q \in [1, \infty]$, $\varphi \in \mathcal{D}_q(\mathbb{R}^d)$, $\sigma > 0$, *and* $m \in \mathbb{N}$. *Then, for every* $f \in \mathrm{conv}_m(\mathcal{V}_\varphi^\sigma)$ *and every* $\varepsilon > 0$, *there is a deep belief network* $p \in \mathsf{DBN}_\varphi(d, m, m+1)$ *such that*

$$\|f - p\|_{L^q} \leqslant \varepsilon.$$

*Proof.* Since $f \in \mathrm{conv}_m(\mathcal{V}_\varphi^\sigma)$, there are by definition of $\triangle_m$ $(\alpha_1, \ldots, \alpha_m) \in \triangle_m$ and $(\mu_1, \ldots, \mu_m) \in (\mathbb{R}^d)^m$ such that

$$f = \sum_{i=1}^m \alpha_i \varphi_{\mu_i, \sigma}.$$

We can think of $\alpha = (\alpha_1, \ldots, \alpha_m)$ as a probability distribution $\tilde{\alpha}$ on $\{0,1\}^m$ by declaring

$$\tilde{\alpha}(h_1) = \begin{cases} \alpha_i, & \text{if } h_1 = e_i, \\ 0, & \text{else}, \end{cases} \qquad h_1 \in \{0,1\}^m,$$

where $(e_i)_j = \delta_{i,j}$, $j = 1, \ldots, m$, is the $i^{\text{th}}$ unit vector.

Let us fix $q \in [1, \infty]$ and $\sigma > 0$. By Proposition 1 there is a $\pi \in \mathsf{B\text{-}RBM}(m, m+1)$ such that

$$\left| \tilde{\alpha}(h_1) - \sum_{h_2 \in \{0,1\}^{m+1}} \pi(h_1, h_2) \right| \leqslant \frac{\varepsilon}{m\sigma \|\varphi\|_{L^q}} \qquad \forall\, h_1 \in \{0,1\}^m. \tag{12}$$

We set

$$p(v \mid h_1) = \begin{cases} \varphi_{\mu_i, \sigma}(v), & h_1 = e_i, \\ 0, & \text{else}, \end{cases}$$

and

$$p(v, h_1, h_2) = p(v \mid h_1)\pi(h_1, h_2) \in \mathsf{DBN}_\varphi(d, m, m+1).$$

This is the desired approximation since

$$\|f - p\|_{L^q} \leqslant \sum_{i=1}^m \left| \alpha_i - \sum_{h_2 \in \{0,1\}^{m+1}} \pi(e_i, h_2) \right| \left\| \varphi_{\mu_i, \sigma} \right\|_{L^q} \leqslant \varepsilon,$$

where we used that $\|\varphi_{\mu, \sigma}\|_{L^q} = \sigma \|\varphi\|_{L^q}$ for each $\mu \in \mathbb{R}^d$ and each $\sigma > 0$. $\qquad\square$

### 4.2 Approximation by Convolution

Let $f \in L^q(\mathbb{R}^d)$, $q \in [1, \infty]$, and $\varphi \in \mathcal{D}(\mathbb{R}^d)$. We denote the *convolution* of $f$ and $\varphi_\sigma$ by

$$(f \star \varphi_\sigma)(x) = \int_{\mathbb{R}^d} f(\mu)\varphi_\sigma(x - \mu)\,d\mu = \int_{\mathbb{R}^d} f(\mu)\varphi_{\mu, \sigma}(x)\,d\mu.$$

Young's convolution inequality, Young (1912), implies that $f \star \varphi_\sigma \in L^q(\mathbb{R}^d)$. In addition, the following approximation result holds, see Appendix A.1 for the proof:

**Proposition 10.** *Let* $\varphi \in \mathcal{D}(\mathbb{R}^d)$. *Then all of the following hold true:*

1. *For each* $q \in [1, \infty)$ *and each* $f \in L^q(\mathbb{R}^d)$, *we have*

$$\lim_{\sigma \downarrow 0} \left\| f - f \star \varphi_\sigma \right\|_{L^q} = 0.$$

2. *If* $f \in L^\infty(\mathbb{R}^d) \cap \mathcal{C}_u(\mathbb{R}^d)$, *then*

$$\lim_{\sigma \downarrow 0} \left\| f - f \star \varphi_\sigma \right\|_{L^\infty} = 0.$$

### 4.3 Approximation Theory in Banach Spaces

The second ingredient needed in the proof of Theorem 3 is an abstract result from the geometric theory of Banach spaces. To formulate it, we need to introduce the following notion: The Rademacher type of a Banach space $(\mathcal{X}, \|\cdot\|_{\mathcal{X}})$ the largest number $\mathsf{t} \geqslant 1$ for which there is a constant $C > 0$ such that, for each $k \in \mathbb{N}$ and each $f_1, \ldots, f_k \in \mathcal{X}$,

$$\mathbb{E}\left[\left\|\sum_{i=1}^{k} \epsilon_i f_i\right\|_{\mathcal{X}}^{\mathsf{t}}\right] \leqslant C \sum_{i=1}^{k} \|f_i\|_{\mathcal{X}}^{\mathsf{t}}$$

holds, where $\epsilon_1, \ldots, \epsilon_k$ are i.i.d. Rademacher random variables, that is, $\mathbb{P}(\epsilon_1 = \pm 1) = \frac{1}{2}$. It can be shown that $\mathsf{t} \leqslant 2$ for every Banach space.

**Example 11.** *The space $L^q(\mathbb{R}^d)$ has Rademacher type $\mathsf{t} = \min(q, 2)$ for $q \in [1, \infty)$. The space $L^{\infty}(\mathbb{R}^d)$ on the other hand has only trivial type $\mathsf{t} = 1$.*

A good reference for the above results on the Rademacher type is (Ledoux & Talagrand, 1991, Section 9.2). The next approximation result and its application to $L^q(\mathbb{R}^d)$ will be important below:

**Proposition 12** (Donahue et al. (1997), Theorem 2.5)**.** *Let $(\mathcal{X}, \|\cdot\|_{\mathcal{X}})$ be a Banach space of Rademacher type $\mathsf{t} \in [1, 2]$. Let $A \subset \mathcal{X}$ and $f \in \overline{\text{conv}}(A)$. Suppose that $\xi = \sup_{g \in A} \|f - g\|_{\mathcal{X}} < \infty$. Then there is a constant $C > 0$ only depending on the Banach space $(\mathcal{X}, \|\cdot\|_{\mathcal{X}})$ such that, for each $m \in \mathbb{N}$, we can find an element $h \in \text{conv}_m(A)$ satisfying*

$$\|f - h\|_{\mathcal{X}} \leqslant \frac{C\xi}{m^{1 - \frac{1}{\mathsf{t}}}}. \tag{13}$$

Notice that the bound (13) is of course trivial for $\mathsf{t} = 1$. Moreover, in Appendix A.2 we provide an example which shows that the convergence rate $m^{\frac{1}{\mathsf{t}} - 1}$ is optimal.

**Corollary 13.** *Let $A \subset L^q(\mathbb{R}^d)$, $1 \leqslant q < \infty$, and suppose that $f \in \overline{\text{conv}}(A)$. If $\xi = \sup_{g \in A} \|f - g\|_{\mathcal{X}} < \infty$, then for all $m \in \mathbb{N}$, there is a $h \in \text{conv}_m(A)$ such that*

$$\|f - h\|_{L^q} \leqslant \frac{\Upsilon_q \xi}{m^{1 - \frac{1}{\min(q,2)}}},$$

*where $\Upsilon_q$ is the constant defined in (8).*

*Proof.* Owing to Example 11 we are in the regime of Proposition 12. The sharp constant $C = \Upsilon_q$ was derived in Haagerup (1981). $\square$

### 4.4 Proof of Theorems 3 and 5

Before giving the technical details of the proofs, let us provide an overview of the strategy:

1. By Proposition 10 we can approximate the density $f \in \mathcal{D}_q(\mathbb{R}^d)$ with $f \star \varphi_\sigma$ up to an error which vanishes as $\sigma \downarrow 0$.

2. Upon showing that $f \star \varphi_\sigma \in \overline{\text{conv}}(\mathcal{V}_\varphi^\sigma)$, Proposition 13 allows us to show that for each $\varepsilon > 0$ and each $m \in \mathbb{N}$, we can pick $\sigma > 0$ such that

$$\inf_{g \in \text{conv}_m(\mathcal{V}_\varphi^\sigma)} \|f - g\|_{L^q} \leqslant \varepsilon + \frac{2\Upsilon_q \|\varphi\|_{L^q}}{m^{1 - \frac{1}{\min(q,2)}}}.$$

3. Finally, we employ Lemma 9 to conclude the desired estimate (9).

**Lemma 14.** *Let $q \in [1, \infty]$, $f \in \mathcal{D}_q(\mathbb{R}^d)$, and $\varphi \in \mathcal{D}(\mathbb{R}^d)$. Then, for each $\sigma > 0$, we have*

$$f \star \varphi_\sigma \in \overline{\text{conv}}(\mathcal{V}_\varphi^\sigma),$$

*with the closure understood with respect to the norm $\|\cdot\|_{L^q}$.*

*Proof.* Let us abbreviate $g = f \star \varphi_\sigma$. We argue by contradiction. Suppose that $g \notin \overline{\text{conv}}(\mathcal{V}^\sigma_\varphi)$. As a consequence of the Hahn-Banach theorem, $g$ is separated from $\overline{\text{conv}}(\mathcal{V}^\sigma_\varphi)$ by a hyperplane. More precisely, there is a continuous linear function $\rho : L^q(\mathbb{R}^d) \to \mathbb{R}$ such that $\rho(h) < \rho(g)$ for all $h \in \overline{\text{conv}}(\mathcal{V}^\sigma_\varphi)$, see (Brezis, 2011, Theorem. 1.7). On the other hand, we however have

$$\rho(g) = \rho\left(\int_{\mathbb{R}^d} f(\mu)\varphi_{\mu,\sigma}\, d\mu\right) = \int_{\mathbb{R}^d} f(\mu)\rho(\varphi_{\mu,\sigma})\, d\mu < \rho(g) \int_{\mathbb{R}^d} f(\mu)\, d\mu = \rho(g),$$

which is the desired contradiction. $\square$

We can now establish the main results of this article:

*Proof of Theorems 3 and 5.* Let us first assume that $q \in (1, \infty)$ and prove the quantitative bound (9). To this end fix $\varepsilon > 0$ and $m \in \mathbb{N}$. We first observe that, by Proposition 10, we can choose $\sigma > 0$ sufficiently small such that $\|f - f \star \varphi_\sigma\|_{L^q} \leqslant \frac{\varepsilon}{2}$. Employing Lemma 14 and Corollary 13 with $A = \mathcal{V}^\sigma_\varphi$, we can find a $g_m \in \text{conv}_m(\mathcal{V}^\sigma_\varphi)$ such that

$$\|f - g_m\|_{L^q} \leqslant \|f - f \star \varphi_\sigma\|_{L^q} + \|f \star \varphi_\sigma - g_m\|_{L^q} \leqslant \frac{\varepsilon}{2} + \frac{\Upsilon_q}{m^{1-\frac{1}{\min(q,2)}}} \sup_{\mu \in \mathbb{R}^d} \|f \star \varphi_\sigma - \varphi_{\mu,\sigma}\|_{L^q}.$$

For the last term we bound

$$\sup_{\mu \in \mathbb{R}^d} \|f \star \varphi_\sigma - \varphi_{\mu,\sigma}\|_{L^q} = \sup_{\mu \in \mathbb{R}^d} \left(\int_{\mathbb{R}^d} \left|\int_{\mathbb{R}^d} f(x)\left(\varphi_\sigma(y-x) - \varphi_\sigma(y-\mu)\right) dx\right|^q dy\right)^{\frac{1}{q}}$$

$$\leqslant \int_{\mathbb{R}^d} f(x) \sup_{\mu \in \mathbb{R}^d} \left(\int_{\mathbb{R}^d} \left|\varphi_\sigma(y-x) - \varphi_\sigma(y-\mu)\right|^q dy\right)^{\frac{1}{q}}$$

$$= \sup_{\mu \in \mathbb{R}^d} \|\varphi - \varphi_{\mu,1}\|_{L^q} \leqslant 2\|\varphi\|_{L^q},$$

whence

$$\|f - g_m\|_{L^q} \leqslant \frac{\varepsilon}{2} + \frac{2\Upsilon_p\|\varphi\|_{L^q}}{m^{1-\frac{1}{\min(q,2)}}}.$$

Finally, Lemma 9 allows us to choose $p \in \text{DBN}_\varphi(d, m, m+1)$ such that $\|g_m - p\|_{L^q} \leqslant \frac{\varepsilon}{2}$. Therefore, we conclude

$$\|f - p\|_{L^q} \leqslant \varepsilon + \frac{2\Upsilon_q\|\varphi\|_{L^q}}{m^{1-\frac{1}{\min(q,2)}}}.$$

Since $\varepsilon > 0$ was arbitrary, the bound (9) follows.

If $q = 1$ or $q = \infty$, we use the fact that

$$\overline{\text{conv}}(A) = \overline{\bigcup_{m \in \mathbb{N}} \text{conv}_m(A)}$$

for any subset $A$ of either $L^1(\mathbb{R}^d)$ or $L^\infty(\mathbb{R}^d)$, respectively. This implies that, for each $\varepsilon > 0$, we can find $m \in \mathbb{N}$ and $g_m \in \text{conv}_m(\mathcal{V}^\sigma_\varphi)$ such that $\|f \star \varphi_\sigma - g_m\|_{L^q} \leqslant \frac{\varepsilon}{3}$. If $q = \infty$, we note that a uniformly continuous and integrable function is always bounded. Hence, in any case we can apply Proposition 10 to find a $\sigma > 0$ for which $\|f - f \star \varphi_\sigma\|_{L^q} \leqslant \frac{\varepsilon}{3}$. Finally employing Lemma 9 as above, there is a $p \in \text{DBN}_\varphi(d, m, m+1)$ such that

$$\|f - p\|_{L^q} \leqslant \|f - f \star \varphi_\sigma\|_{L^q} + \|f \star \varphi_\sigma - g_m\|_{L^q} + \|g_m - p\|_{L^q} \leqslant \varepsilon. \qquad \square$$

### 4.5 Kullback-Leibler Approximation on Compacts

Let us begin by bounding the Kullback-Leibler divergence in terms of the $L^2$-norm:

**Lemma 15** (Zeevi & Meir (1997), Lemma 3.3). *Let $\Omega \subset \mathbb{R}^d$, $f : \Omega \to \mathbb{R}_+$, and $g : \mathbb{R}^d \to \mathbb{R}_+$ be probability densities. If there is an $\eta > 0$ such that both $f, g \geqslant \eta$ on $\Omega$, then*

$$\text{KL}(f\|g) \leqslant \frac{1}{\eta}\|f - g\|^2_{L^2(\Omega)}.$$

*Proof.* We use Jensen's inequality and the elementary fact $\log x \leqslant x - 1$, $x > 0$, to obtain

$$
\mathrm{KL}(f\|g) = \int_\Omega \log\left(\frac{f(x)}{g(x)}\right) f(x)\, dx \leqslant \log\left(\int_\Omega \frac{f(x)^2}{g(x)}\, dx\right)
$$
$$
\leqslant \int_\Omega \frac{f(x)^2}{g(x)}\, dx - 1 = \int_\Omega \frac{(f(x) - g(x))^2}{g(x)}\, dx \leqslant \frac{1}{\eta}\|f - g\|_{L^2}^2.
$$

$\square$

Finally, we can prove the approximation bound in Kullback-Leibler divergence:

*Proof of Corollary 7.* Extending the target density $f$ by zero on $\mathbb{R}^d \setminus \Omega$, the corollary follows from Theorem 3 upon showing that, for each $m \in \mathbb{N}$, we can choose the approximation $p \in \mathsf{DBN}_\varphi(d, m, m+1)$ in such a way that $p \geqslant \frac{\eta}{2}$ on $\Omega$.

To see this, we notice that $f$ is uniformly continuous since $\Omega$ is compact. Hence, Theorem 5 allows us to pick an $M \in \mathbb{N}$ such that, for each $m \geqslant M$, there is a $p_m \in \mathsf{DBN}_\varphi(d, m, m+1)$ with $\|f - p_m\|_{L^\infty} \leqslant \frac{\eta}{2}$. In particular, each of these DBNs satisfies $p_m \geqslant \frac{\eta}{2}$ on $\Omega$. Consequently, by Lemma 15 we obtain

$$
\inf_{p \in \mathsf{DBN}_\varphi(d, m, m+1)} \mathrm{KL}(f\|p) \leqslant \frac{8\|\varphi\|_{L^2}^2}{\eta m} \qquad \forall m \geqslant M. \tag{14}
$$

A crude upper bound on $\inf_{p \in \mathsf{DBN}_\varphi(d,m,m+1)} \mathrm{KL}(f\|p)$ for $m < M$ can be obtained choosing both zero weights and biases in (1) as well as $p(v \mid h_1) = \varphi$ for each $h_1 \in \{0, 1\}^m$ in (3). Hence, the visible units of the DBN have density $\varphi$. This gives

$$
\inf_{p \in \mathsf{DBN}_\varphi(d, m, m+1)} \mathrm{KL}(f\|p) \leqslant \mathrm{KL}(f\|\varphi) \leqslant \frac{1}{\eta}\|f - \varphi\|_{L^2(\Omega)}^2 \qquad \forall m = 1, \ldots, M-1, \tag{15}
$$

again by Lemma 15. Finally, combining (14) and (15) we get the required estimate:

$$
\inf_{p \in \mathsf{DBN}_\varphi(d, m, m+1)} \mathrm{KL}(f\|p) \leqslant \frac{M}{\eta m}\left(8\|\varphi\|_{L^2}^2 + \|f - \varphi\|_{L^2(\Omega)}^2\right). \qquad \square
$$

**Remark 16.** *Our strategy of the proof of the Kullback-Leibler approximation in Corollary 7 through Lemma 15 differs from the one employed in (Krause et al., 2013, Theorem 7). There, the authors built on the results of Li & Barron (1999) and in the course of their argument claim that the following statement holds true:*

Let $f_m, f : \Omega \to \mathbb{R}_+$, $m \in \mathbb{N}$, be probability densities on a compact set $\Omega \subset \mathbb{R}^d$ with $f_m, f \geqslant \eta > 0$. If $\mathrm{KL}(f\|f_m) \to 0$ as $m \to \infty$, then $f_m \to f$ in the norm $\|\cdot\|_{L^\infty}$.

*This, however, does not hold as we illustrate by a simple counterexample presented in Appendix A.3.*

## 5 CONCLUSION

We investigated the approximation capabilities of deep belief networks with two binary hidden layers of sizes $m$ and $m+1$, respectively, and real-valued visible units. We showed that, under minimal regularity requirements on the parental density $\varphi$ as well as the target density $f$, these networks are universal approximators in the strong $L^q$ and Kullback-Leibler distances as $m \to \infty$. Moreover, we gave sharp quantitative bounds on the approximation error. We emphasize that the convergence rate in the number of hidden units is independent of the choice of the parental density.

Our results apply to virtually all practically relevant examples thereby theoretically underpinning the tremendous empirical success of DBN architectures we have seen over the last couple of years. As we alluded to in Remark 4, the frequently made choice of a Gaussian parental density does not provide the theoretically optimal DBN approximation of a given target density. Since, in practice, the choice of parental density cannot solely be determined from an approximation standpoint, but also the difficulty of the training of the resulting networks needs to be considered, it is interesting to further empirically study the choice of parental density on both artificial and real-world datasets.

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

## A  DETAILS OF THE MATHEMATICAL RESULTS

This appendix provides further details of the mathematical results used in the main text. More specifically, we provide

1. the proof of Proposition 10,

2. a detailed proof of Proposition 12 for Hilbert spaces as well as an example showing that its approximation rate is optimal in general, and

3. the construction of an explicit counterexample to the statement discussed in Remark 16.

### A.1  PROOF OF PROPOSITION 10

*Proof.* Item 1 is well known, see e.g. (Folland, 1999, Theorem 8.14). For 2 fix $\varepsilon > 0$. By uniform continuity of $f$, we can find a $\delta > 0$ such that

$$\sup_{|\mu|\leqslant\delta} \left|f(x) - f(x-\mu)\right| \leqslant \frac{\varepsilon}{2} \qquad \forall\, x \in \mathbb{R}^d. \tag{16}$$

In particular, we obtain

$$\left|f(x) - \left(f \star \varphi_\sigma\right)(x)\right|$$

$$\leqslant \int_{\mathbb{R}^d} \varphi_\sigma(\mu)\left|f(x) - f(x-\mu)\right| d\mu$$

$$\leqslant \int_{\{|\mu|>\delta\}} \varphi_\sigma(\mu)\left|f(x) - f(x-\mu)\right| d\mu + \int_{\{|\mu|\leqslant\delta\}} \varphi_\sigma(\mu)\left|f(x) - f(x-\mu)\right| d\mu$$

$$\leqslant 2\|f\|_{L^\infty} \int_{\{|\mu|>\delta\}} \varphi_\sigma(\mu)\, d\mu + \frac{\varepsilon}{2},$$

where we applied the uniform continuity estimate (16) to the second integral. Since

$$\int_{\{|\mu|>\delta\}} \varphi_\sigma(\mu)\, d\mu = \int_{\left\{|\mu|>\frac{\delta}{\sigma}\right\}} \varphi(\mu)\, d\mu \to 0 \qquad \text{as } \sigma \downarrow 0,$$

we can choose $\sigma_0 > 0$ such that $\left\|f - \left(f \star \varphi_\sigma\right)\right\|_{L^\infty} \leqslant \varepsilon$ for all $\sigma \in (0, \sigma_0)$. This completes the proof.

$\square$

### A.2  DETAILS ON PROPOSITION 12

While the proof of Proposition 12 for a general Banach space is rather technical, we find it instructive to present the simplified argument for a *Hilbert* space. Our proof is inspired by Jones (1992), see also Barron (1994).

**Proposition 17.** *Let $\left(\mathcal{X}, \|\cdot\|_\mathcal{X}\right)$ be a Hilbert space. Let $A \subset \mathcal{X}$ and $f \in \overline{\mathrm{conv}}(A)$. Suppose that $\xi = \sup_{g\in A} \|f - g\|_\mathcal{X} < \infty$. Then, for each $m \in \mathbb{N}$, we can find an element $g \in \mathrm{conv}_m(A)$ satisfying*

$$\|f - g\|_\mathcal{X} \leqslant \frac{\xi}{\sqrt{m}}. \tag{17}$$

*Proof.* We proceed by induction on $m \in \mathbb{N}$. The base $m = 1$ is trivial, so we can assume that the statement holds for $m \geqslant 1$. Let us declare

$$\Xi_{m+1} = \inf_{g\in\mathrm{conv}_{m+1}(A)} \|f - g\|_\mathcal{X}.$$

By the induction hypothesis, we may assume that $\Xi_m \leqslant \frac{\xi}{\sqrt{m}}$ and we can find $h \in \operatorname{conv}_m(A)$ attaining this bound. Consequently, we get

$$
\begin{aligned}
\Xi_{m+1}^2 &\leqslant \inf_{\substack{\lambda \in [0,1] \\ g \in A}} \left\| \lambda(f-g) + (1-\lambda)(f-h) \right\|_{\mathcal{X}}^2 \\
&= \inf_{\substack{\lambda \in [0,1] \\ g \in A}} \left[ \lambda^2 \|f-g\|_{\mathcal{X}}^2 + 2\lambda(1-\lambda) \langle f-g, f-h \rangle_{\mathcal{X}} + (1-\lambda)^2 \|f-h\|_{\mathcal{X}}^2 \right] \qquad (18) \\
&\leqslant \inf_{\lambda \in [0,1]} \left[ \lambda^2 \xi^2 + 2\lambda(1-\lambda) \inf_{g \in A} \langle f-g, f-h \rangle_{\mathcal{X}} + (1-\lambda)^2 \Xi_m^2 \right].
\end{aligned}
$$

We claim that

$$
\inf_{g \in A} \langle f-g, f-h \rangle_{\mathcal{X}} = 0. \qquad (19)
$$

To see this, let us fix an $\varepsilon > 0$ and observe that, since $f \in \overline{\operatorname{conv}}(A)$, the Cauchy-Schwarz inequality implies that there must be a finite convex combination of elements in $A$ satisfying

$$
\sum_{i=1}^k \alpha_i \langle f - a_i, f - h \rangle = \left\langle f - \sum_{i=1}^k \alpha_i a_i, f - h \right\rangle \leqslant \varepsilon.
$$

In particular, the inequality $\langle f - a_i, f - h \rangle \leqslant \varepsilon$ holds for at least one vector $a_i \in A$. Since $\varepsilon > 0$ was arbitrary, we have established (19).

Inserting (19) in (18), we arrive at

$$
\Xi_{m+1}^2 \leqslant \inf_{\lambda \in [0,1]} \left[ \lambda^2 \xi^2 + (1-\lambda)^2 \Xi_m^2 \right] \leqslant \frac{\xi^2 \Xi_m^2}{\xi^2 + \Xi_m^2},
$$

where the last step follows by chosing $\lambda = \frac{\Xi_m^2}{\Xi_m^2 + \xi^2} \in [0,1]$. Finally, recalling the induction hypothesis $\Xi_m \leqslant \frac{\xi}{\sqrt{m}}$, we conclude

$$
\Xi_{m+1}^2 \leqslant \xi^2 \frac{\frac{\xi^2}{m}}{\xi^2 + \frac{\xi^2}{m}} = \frac{\xi^2}{m+1}.
$$

This establishes (17) for $m+1$ and the induction is complete. $\qquad \square$

Returning to the original statement of Proposition 12 for a general Banach space, the next example shows that its convergence rate is optimal in general:

**Example 18.** *For $p \in (1,2]$ let us consider the Banach space $\ell^p(\mathbb{R})$ of p-summable real-valued sequences, that is, $(a_n)_{n \in \mathbb{N}} \subset \mathbb{R}$ belongs to $\ell^p(\mathbb{R})$ iff*

$$
\|a\|_{\ell^p} = \left( \sum_{n=1}^\infty |a_n|^p \right)^{\frac{1}{p}} < \infty.
$$

*It can be shown that this Banach space has Rademacher type $\mathfrak{t} = p$. Let $A$ be the set formed of the standard basis vectors:*

$$
A = \big\{ (1,0,0,0,\dots), (0,1,0,0,\dots), (0,0,1,0,\dots), \dots \big\}.
$$

*Choosing $f \equiv 0$, we find that*

$$
\inf_{h \in \operatorname{conv}_m(A)} \|f - h\|_{\ell^p} = \inf_{(\alpha_1,\dots,\alpha_m) \in \triangle_m} \left( \sum_{i=1}^m \alpha_i^p \right)^{\frac{1}{p}}.
$$

*The optimum on the right-hand side is attained by choosing $\alpha_1 = \cdots = \alpha_m = \frac{1}{m}$ so that*

$$
\inf_{h \in \operatorname{conv}_m(A)} \|f - h\|_{\ell^p} = \frac{1}{m^{1 - \frac{1}{p}}} = \frac{1}{m^{1 - \frac{1}{\mathfrak{t}}}}.
$$

### A.3 CONSTRUCTION OF THE COUNTEREXAMPLE IN REMARK 16

Let $\Omega = [0, 1]$ and consider the sequence of probability densities given by

$$f_m(x) = C_m \left( 1 \wedge \left( mx + \frac{1}{2} \right) \right), \qquad m \in \mathbb{N},$$

where $C_m = (1 - 1/(8m))^{-1}$ is chosen such that $\int_0^1 f_m(x) \, dx = 1$. Then we have $f_m(x) \to \mathbb{1}_{[0,1]}(x) = f(x)$ pointwise on $(0, 1]$. On the other hand, it holds that

$$\sup_{x \in [0,1]} \left| f_m(x) - f(x) \right| = \left| f_m(0) - f(0) \right| = \frac{1}{2} \qquad \forall m \in \mathbb{N}.$$

Consequently, $f_m$ does not converge uniformly to $f$.

Nevertheless, it is straight-forward to check that $\left\| f_m - \mathbb{1}_{[0,1]} \right\|_{L^2} \to 0$ and since $f_m, f \geqslant 1/2$ on $\Omega$, we have $\mathrm{KL}(f_m \| f) \to 0$ as $m \to \infty$ by Lemma 15.

