# OpenReview forum: "Quantitative Universal Approximation Bounds for Deep Belief Networks"
_ICLR.cc/2023/Conference — Submitted to ICLR 2023_

### Official Review · Reviewer_F3UE · 2022-10-21

**Confidence:** 4
**Correctness:** 4
**Technical Novelty And Significance:** 3
**Empirical Novelty And Significance:** Not applicable
**Recommendation:** 8

**Clarity, Quality, Novelty And Reproducibility:**

The paper is well-written and easy to follow. I believe the writing can be improved by moving some less important math to appendix and add more explanantion/discussion to the main-text.

**Strength And Weaknesses:**

Strength:

The results are new and solid, generalizing previous results for the discrete case to the continuous case.

The proof idea is simple yet effective, through which this paper solves the targeted problem in a clean way.

Weaknesses:

The result is well-expected due to numerous analogical universal approxiamtion results for neural networks, and similar techniques have appeared in that literature.

What's the dependence of $M$ on $\epsilon$ in theorem 3?

The analysis is based on some technical propositions from previous works without proof. Adding more explanation/proof idea/easy example to these introduced propositions would make this paper more readable. For instance, proposition 11 could benefit from explaining the special case of $L^2$, which admits a short proof because it's a Hilbert space.

**Summary Of The Paper:**

This paper proves univeral approximation theorems for deep belief networks when the targeted distribution is continuous, with quantitative bounds for $L^p$ space with $p>1$. The proof idea is to make use of the efficient approximation of convex hull, each item is represented in the first layer, while the second layer is used to approximate the convex combination utilizing previous results for the discrete case.

**Summary Of The Review:**

The results are new and solid, written in a mathematically clean way. I recommend accept.

---

> ### Author Response · Authors · 2022-11-12
> **Thank you for your review!**
>
> We are very grateful to the reviewer for their time spent in reading our paper and their valuable comments. As we detail below, we addressed all of their comments in our revision:
>
> ### What's the dependence of $M$ on $\varepsilon$ in Theorem 3?
> We agree that Theorem 3 was formulated in an obfuscated way. To mitigate this, we moved the quantitative bound to the beginning of the theorem and clarified that we only have a weaker qualitative statement for $q=1$.
>
> ### Add more explanations to the proofs and move technical parts to Appendix
> You’re right that some of our proofs lack some details hard to fill in for non-expert readers. This was also pointed out by reviewer atRJ. We therefore decided to add a technical appendix containing the following:
> We moved Proposition 10 to the Appendix and added additional details to its proof.
> We give a detailed proof of Proposition 12 for the special case of a Hilbert space (see Proposition 17 in the Appendix).
> We provide an example illustrating that the approximation of Proposition 12 is optimal in general.
> We moved the counterexample in Remark 16 to the Appendix and provided a more thorough explanation.
>
> If there are any points from your review, which we missed to address, please do not hesitate to reach out to us.

---

### Official Review · Reviewer_F7vv · 2022-10-24

**Confidence:** 2
**Correctness:** 4
**Technical Novelty And Significance:** 3
**Empirical Novelty And Significance:** 3
**Recommendation:** 6

**Clarity, Quality, Novelty And Reproducibility:**

This paper is well-written. The result of the theoretical analysis in this paper is novel and the obtained bounds of the approximation error of DBNs in the case of two hidden layers of size m and m+1, respectively.

**Strength And Weaknesses:**

Strengths:

1. This is a solid theoretical work on the analysis of approximation error of DBN, which has been an open problem.

2. Two metrics, L^q-norm and KL deivergence, are considered.

Weaknesses:

1. It seems that the main results of this manuscript are restricted only to the case of DBNs with two hidden layers of size m and m+1 respectively. Can the results still hold for general size m and n for two layers? If not, why?

2. The authors stated that, e.g., in the abstract,  "we establish a sharp quantitative bounds on the approximation error in terms of the number of hidden units ". How to verify the sharpness of the obtained bounds? I did not see any supportive results to show that the obtained bounds are sharp. Can the authors add more details or explanations on this point? I would be very nice if some empirical evidence is provided.

**Summary Of The Paper:**

This manuscript investigates the approximation error of classic deep belief networks (DBNs), in particular DBNs with two hidden layers of size m and m+1, respectively. It is demonstrated that, under both  L^q-norm and Kullback-Leibler divergence, DBNs are universal approximators. Moreover,  as claimed by the author, sharp bounds are obtained for the approximation error.

**Summary Of The Review:**

This manuscript is theoretically studied the approximation error of DBNs with   two binary hidden layers of sizes m and m + 1, respectively. In particular, under both L^q and Kullback-Leibler divergence metrics,  DBNs are proven to be universal approximators and the bounds of the approximation errors have been obtained.  It is a solid work with theoretical contributions. My main concern lies in the restriction of the sizes of the hidden layers to be m and m + 1, as well as a lack of (empirical) evidence of the sharpness of the obtained bounds.

---

> ### Author Response · Authors · 2022-11-12
> **Thank you for your review!**
>
> We are very grateful to the reviewer for their time spent in reading our paper and their valuable comments. As we detail below, we addressed all of their comments in our revision:
>
> ### Can the results still hold for general size $m$ and $n$ for two layers? If not, why?
> The result holds for a second layer of any size greater than or equal to $m+1$. This is because the RBM approximation result in Proposition 1 holds for any number of nodes in the hidden RBM layer (that is, the second hidden layer of the DBN) greater than m. We believe that this result of Le Roux and Bengio is “optimal” in the sense that one can find a discrete probability distribution on $\{1,\dots, m\}$ which can’t be arbitrarily well approximated by an RBM with $m$ visible and $m$ hidden layers. Therefore, the result of Theorem wouldn’t hold for a second hidden layer of size less than $m+1$ since the coefficients in the convex combination couldn’t be approximated by arbitrary accuracy, as we exploited in Lemma 9.
> An interesting question for future work is whether the approximation bound in Theorem 3 can be improved when adding more hidden layers. A partial result in this direction has been obtained by Jalali et al. (NeurIPS 2019). We added a short discussion in Remark 8, but left a thorough investigation to future work.
>
> ### How to verify the sharpness of the obtained bounds?
> The sharpness of the obtained bounds follows in an indirect manner from the optimal rate and constant in Corollary 13 and the sharpness of our arguments ultimately proving Theorem 3. We clarified this by adding an explicit example for the optimality of the rate in Proposition 12 in the Appendix, see Example 18. Nevertheless, we strongly agree with the reviewer’s suggestion of illustrating the convergence rate in Theorem 3 by an empirical study of different choices of parental densities for DBNs. While we cannot include such a study as part of the present article due to the page limit, we are preparing an empirical companion article which will include empirical evidence for sharpness of the estimate of Theorem 3.
>
> If there are any points from your review, which we missed to address, please do not hesitate to reach out to us.

---

### Official Review · Reviewer_V3vX · 2022-10-25

**Confidence:** 4
**Correctness:** 4
**Technical Novelty And Significance:** 1
**Empirical Novelty And Significance:** Not applicable
**Recommendation:** 3

**Clarity, Quality, Novelty And Reproducibility:**

The paper does not give a clear introduction for deep belief networks. The presentation can be improved. The mathematical results are not so deep, without using any regularity of the approximated function.

**Strength And Weaknesses:**

The topic of approximation by deep belief networks is interesting. Rates of convergence are given by applying the Rademacher index of the L_q spaces. There have been many results in the literature of distribution regression, on learning of probability density functions. The authors do not provide any comparisons with the existing methods for approximating probability density functions. Using convolutions for approximations is fine, but regularity of the approximated function is not used.

**Summary Of The Paper:**

The authors consider approximation of probability density functions by deep belief networks with binary hidden units. Convergence and rates of approximation are provided with respect to the L_q norm and K-L divergence.

**Summary Of The Review:**

The topic of approximation by deep belief networks is interesting. Rates of convergence given by applying the Rademacher index of the L_q spaces might have applications in some other problems. But a large literature of distribution regression is missing in the paper. Using convolutions for approximations without involving regularity of the approximated function is rather shallow.

---

> ### Author Response · Authors · 2022-11-05
> **Request of brief clarification**
>
> We thank the reviewer for their insightful comments. It seems to us that the absence of an explicit use of the regularity of the approximated density in our arguments is a major concern of the reviewer. Before addressing this and the other points in a subsequent comment, we would like to kindly ask for further clarification with regards to the regularity of the approximated function. In particular, we would like to know which kind of regularity the reviewer has in mind (apart from the already used $L^q$-integrability requirements).

---

> ### Author Response · Authors · 2022-11-12
> **Thank you for your review!**
>
> We thank the reviewer for their insightful comments and proposed changes to improve the article. We want to address the two main points as follows:
>
> ### Comparison to other approximations of probability densities
> We thank the reviewer for pointing out that references to other approximation methods in the context of probability distributions would be beneficial. We added several references to classical and modern monographs on mixture models, recent developments in the context of generative networks as well as some references to general quantile regression techniques.
>
> ### Too short introduction to deep belief networks
> We thank the reviewer for pointing out that additional material on top of the references in the introduction would be beneficial. We added additional context regarding the first use of DBNs as well as a sentence on their training through contrastive divergence. For a more complete review we reference the reader to a survey article on RBMs and DBNs. In case there are any additional points on DBN that you would see beneficial to the article’s readability we are more than happy to incorporate them into our article.
>
> ### Deepness of mathematical results
> We agree that qualitative statements could have been expected to hold true given the vast literature on approximation capabilities of deep neural networks, we respectfully disagree with the reviewer’s assessment of the novelty of our results. In particular, we think that the quantitative rate of convergence in Theorem 3 is a strong theoretical contribution which, to our best knowledge, hasn’t seen any comparable investigation in the literature yet. This is backed up by the assessments of reviewers heng, F7vv, and F3UE.
>
> If there are any points from your review, which we missed to address, please do not hesitate to reach out to us.

---

### Official Review · Reviewer_heng · 2022-10-26

**Confidence:** 4
**Correctness:** 4
**Technical Novelty And Significance:** 3
**Empirical Novelty And Significance:** Not applicable
**Recommendation:** 8

**Clarity, Quality, Novelty And Reproducibility:**

**Clarity**

The paper is well written and is easy to read and understand.

**Novelty and Significance**

I think the error rates presented are new and are of theoretical significance.


**Strength And Weaknesses:**

**Strengths**
---

1) I think the major strength of the paper are the error rates that it presents. In particular, I think even for Gaussian Mixture I dont the optimal error rate is known.

2) I think writing is clear and proofs are easy to follow and use mostly standard analysis tools.

**Weaknesses**
---

1) I think a discussion to prior theoretical work on approximations by mixture models is missing. Specifically in the case when we think of the special cases of mixtures of Gaussians and mixtures of exponential families. Significant prior work has been done. In this regard, I think the universality is known (please correct me if I am wrong). However, the strength of the this paper are the error rates.

**Summary Of The Paper:**

This paper looks at the approximation capabilities of a DBN with 2 layers in which the first layer is a has binary states (both hidden and visible) and the second layer has continuous visible state. Further, it is assumed that the conditional distributions for the last layer all come from the same parental distribution (that is, they are just shifts and rescalings of each other). Under such an assumption, we can think of the DBN as mixture model (such as a Gaussian Mixture model with $\sigma I$ as the covariances when the parental distribution is Gaussian).

In this set up the paper shows that as the number of hidden nodes goes to infinity, the model can approximate arbitrarily well any measure that is absolutely continuous with respect to Lebesgue measure (i.e. measures that have distributions with respect to the Lebesgue measure.) That is, as the number of mixtures factors increases we can approximately any density.

In some cases the paper provides error rates.



**Summary Of The Review:**

In summary, I think this is a well written paper and with interesting and new results.

---

> ### Author Response · Authors · 2022-11-12
> **Thank you for your review!**
>
> We thank the reviewer for their insightful comments and the proposed changes improving the article. We want to address the two main points as follows:
>
> ### Literature on mixture models
> We thank the reviewer for pointing out that additional references to classical mixture models results would be beneficial. We added a discussion and several references to classical and modern monographs on mixture models as part of the Introduction.
>
> ### References for universality of mixture models
> We thank the reviewer for pointing out that the article would benefit from additional references to universal approximation properties of mixture models. We added a discussion of related works and referred to two survey papers concerning the universal approximation property. We agree that the qualitative statement is known for Gaussian and exponential family mixtures and that the main strength of our generalisation to arbitrary probability densities lies in the explicit rate, which, to our best knowledge, does not exist in the literature yet.
>
> If there are any points from your review, which we missed to address, please do not hesitate to reach out to us.

---

> > ### Comment · Reviewer_heng · 2022-11-18
> > **Thanks for the clarification**
> >
> > Thanks you have addressed my questions.

---

### Official Review · Reviewer_atRJ · 2022-11-01

**Confidence:** 3
**Correctness:** 4
**Technical Novelty And Significance:** 4
**Empirical Novelty And Significance:** Not applicable
**Recommendation:** 6

**Clarity, Quality, Novelty And Reproducibility:**

Clarity:  Overall, I found the motivation and the presentation of the theorems clear (though I had more difficulty following the proof details due to lack of expertise).  Still, I have a few comments:

(1) L^q(R^d) is used in the intro before being defined.
(2) At the beginning of section 1, RBMs are described as both planar and fully connected (these statements are antithetical).
(3) The notation in Theorem 3 isn't consistent with Proposition 1 -- don't you need to sum over the hidden variables?

Quality:  I'm not able to assess overall quality as I am not able to validate all of the proof details (and the proofs are a bit terse for a non-expert like me to follow completely).

Novelty:  The work does appear to make novel contributions over existing works, and the authors are careful to point out exactly where those contributions are.

Reproducibility: n/a

**Strength And Weaknesses:**

Strengths:  Fills gaps in the theoretical literature on what densities can be approximated by DBNs.

Weaknesses:  Proofs are a bit terse (as is necessitated by space).  Doesn't contain any experimental evaluation, but I don't think that is necessary for papers of this type.

**Summary Of The Paper:**

A purely theoretical contribution in which the authors present a series of approximation results describing which densities can be well-approximated by DBNs.  Additionally, they provide quantitative error bounds for both Lp based norms and the KL divergence.

**Summary Of The Review:**

An interesting theoretical contribution that precisely describes classes of probability densities that can be well approximated by DBNs.

---

> ### Author Response · Authors · 2022-11-12
> **Thank you for your review!**
>
> We thank the reviewer for their insightful comments and their proposed changes to improve the article. We want to address the four main points as follows:
>
> ### The definition of $L^q(\mathbb{R}^d)$ is after its first usage
> We thank the reviewer for spotting this issue. We moved the definition directly after its first usage.
>
> ### Definition of RBMs as planar graphs
> We thank the reviewer for spotting this issue. We removed the word “planar” from the definition of an RBM.
>
> ### Different notation for evaluation of RBMs and DBNs
> We thank the reviewer for spotting this difference in notation. We clarified our notation directly after the definition of a DBN by stating that whenever we use $p$ as a function on $\mathbb{R}^{|V|}$, we implicitly sum over all hidden variables.
>
> ### Terse proofs
> We thank the reviewer for pointing out that some proofs would benefit from additional details. We added an appendix providing more detailed proofs as well as additional examples illustrating the statements of the abstract mathematical results. More concretely, we moved the proof of Proposition 10 to the Appendix and added details to its presentation. We also discuss the counterexample of Remark 16 in more detail in the Appendix. Last but not least, we provided a full proof of Proposition 12 in the Hilbert space case.
>
> If there are any points from your review, which we missed to address, please do not hesitate to reach out to us.

---

### Author Response · Authors · 2022-11-13
**Revision overview**

We thank the reviewers for their careful reading of our manuscript and their insightful feedback. As we detail in our comments to each of the reviewers, we have incorporated all of their points in our revision. A high-level overview of the changes of the manuscript is below:

* P1. We moved up the definitions of the $L^q$ spaces to their first appearance.
* P2. We added additional references to classical monographs on mixture models, approximation of probability distributions as well as survey articles on universal approximation properties for mixture models.
* P.2. We removed the term “planar” in the definition of an RBM as spotted by reviewer atRJ
* P3-4. We added a convention detailing how we use a DBN as a function by summing over its hidden states. Moreover, we added additional details on how DBN are trained and their history.
* P4. We added an additional reference to a classical monograph on the exponential family to Remark 4.
P4. We added the exponent $\frac{1}{q}$ to the Gamma function on the right-hand side of equation (8). This was a typo in the initial version
* P4. The statement of Theorem 3 is improved based on the comment by reviewer F3UE.
* P5. We added a remark discussing the situation for deeper DBNs.
* P7. We moved the proof of Proposition 10 to the appendix to allow for a more detailed treatment.
* P7. We corrected a typo in the statement of Proposition 12 (changed g to h in the last line of the statement).
* P.7. We changed “Notice that the bound (13) is of course useless for $\mathfrak{t} = 1$” to “Notice that the bound (13) is of course trivial for $\mathfrak{t} = 1$”.
* We added an Appendix containing
  * the proof of Proposition 10,
  * a detailed proof of Proposition 12 for Hilbert spaces as well as an example showing that its approximation rate is optimal in general, and
  * the construction of an explicit counterexample to the statement discussed in Remark 16.


We are happy to provide further clarification if any of the reviewers’ concerns are not answered.

---

### Decision · Program_Chairs · 2023-01-20

**Decision:**

Reject

**Justification For Why Not Higher Score:**

The paper makes promising and interesting contributions to the theoretical understanding of DBNs with continuous visible units and I appreciate that the authors made several improvements to the manuscript following the discussions with the reviewers. Still, I find that the article could be further developed in some ways that would have made it a stronger contribution.

* The article includes many references (particularly after the rebuttal), but misses relevant works such as [1], which gave approximation error bounds for Gaussian Bernoulli RBMs and binary Gaussian DBNs and should be referenced and discussed. The submission has some differences in that it considers approximation in Lp norms and other types of parental distributions. Nonetheless this does to some degree diminish the novelty value of the submission. I appreciate that [1] is relatively recent, but note that it appeared almost 4 months before the ICLR 2023 deadline.
* The contribution would have been stronger by adding a discussion of the comparison of different parental distributions, which the authors identified as an interesting direction but deferred to future work.
* The contribution would have been stronger by adding an empirical illustration of the rates, which the authors again deferred to future work.
* While the results remain valid for deeper architectures, they do not clarify particular benefits of deeper architectures. Similarly, they have limited explanatory power in comparing DBNs with plain mixture models.

Minor points

* If not the title, then at least the abstract should more explicitly indicate the focus on DBNs with two hidden layers and continuous probability distributions.
* The main results appear to be restricted to upper bounds and it would be good to more explicitly and clearly indicate which bounds are sharp and in what sense. The comment after Proposition 12 seems to pertain to mixtures.


[1] Linyan Gu, Lihua Yang, Feng Zhou, Approximation properties of Gaussian-binary restricted Boltzmann machines and Gaussian-binary deep belief networks, Neural Networks, Volume 153, 2022, Pages 49-63.



**Justification For Why Not Lower Score:**

NA

**Metareview: Summary, Strengths And Weaknesses:**

The article investigates the approximation power of deep belief networks with continuous visible units.

* Strengths are clarity of writing and novelty of the presented approximation error bounds for mixtures of continuous distributions in the context of shallow DBNs.
* Weaknesses are the restriction to DBN architectures with two hidden layers and limited description beyond mixtures, moderate innovation in terms of the proof techniques, and missed opportunity to further explore different types of visible distributions (called parental distributions in the submission).

**Summary Of Ac-Reviewer Meeting:**

The article has mixed overall recommendations and a borderline overall rating.

The AC-reviewer meeting highlighted that the paper is very well written and includes interesting results, particularly the error bounds on approximations by mixtures of Gaussians. Reviewers pointed out that the main derivations are well written and easy to follow, but at the same time it was mentioned that the proofs are more or less straight forward from existing results and that several items have been deferred to future work, specifically the illustration of different types of continuous visible units and the illustration of the convergence rates in theorem 3 by empirical study. The authors indicate that some of these were not added due to page constraints and that they are preparing a companion article. I believe that this type of supporting material would fit very well in an appendix and that it would have made this a stronger contribution.

One of the reviewers with 8 rating indicated that 7 (not available in the system) would more accurately reflect their assessment of the paper.